# Comparative Study of Plan Robustness for Breast Radiotherapy: Volumetric Modulated Arc Therapy Plans with Robust Optimization versus Manual Flash Approach

**DOI:** 10.3390/diagnostics13223395

**Published:** 2023-11-07

**Authors:** Ray C. K. Chan, Curtise K. C. Ng, Rico H. M. Hung, Yoyo T. Y. Li, Yuki T. Y. Tam, Blossom Y. L. Wong, Jacky C. K. Yu, Vincent W. S. Leung

**Affiliations:** 1Department of Health Technology and Informatics, Faculty of Health and Social Sciences, The Hong Kong Polytechnic University, Hong Kong SAR, China; 19060221d@connect.polyu.hk (R.C.K.C.); 19065666d@connect.polyu.hk (Y.T.Y.L.); 19050222d@connect.polyu.hk (Y.T.Y.T.); 18052836d@connect.polyu.hk (B.Y.L.W.); 19053802d@connect.polyu.hk (J.C.K.Y.); 2Curtin Medical School, Curtin University, GPO Box U1987, Perth, WA 6845, Australia; curtise.ng@curtin.edu.au; 3Curtin Health Innovation Research Institute (CHIRI), Faculty of Health Sciences, Curtin University, GPO Box U1987, Perth, WA 6845, Australia; 4Department of Clinical Oncology, Pamela Youde Nethersole Eastern Hospital, Hong Kong SAR, China; ricohung@connect.hku.hk

**Keywords:** cancer, chest wall displacement, dosimetry, error, organs at risk, radiation dose, radiation therapy, skin flash, target volume, uncertainty

## Abstract

A previous study investigated robustness of manual flash (MF) and robust optimized (RO) volumetric modulated arc therapy plans for breast radiotherapy based on five patients in 2020 and indicated that the RO was more robust than the MF, although the MF is still current standard practice. The purpose of this study was to compare their plan robustness in terms of dose variation to clinical target volume (CTV) and organs at risk (OARs) based on a larger sample size. This was a retrospective study involving 34 female patients. Their plan robustness was evaluated based on measured volume/dose difference between nominal and worst scenarios (ΔV/ΔD) for each CTV and OARs parameter, with a smaller difference representing greater robustness. Paired sample *t*-test was used to compare their robustness values. All parameters (except CTV ΔD_98%_) of the RO approach had smaller ΔV/ΔD values than those of the MF. Also, the RO approach had statistically significantly smaller ΔV/ΔD values (*p* < 0.001–0.012) for all CTV parameters except the CTV ΔV_95%_ and ΔD_98%_ and heart ΔD_mean_. This study’s results confirm that the RO approach was more robust than the MF in general. Although both techniques were able to generate clinically acceptable plans for breast radiotherapy, the RO could potentially improve workflow efficiency due to its simpler planning process.

## 1. Introduction

Breast cancer is the most common cancer globally. Breast radiotherapy plays an important role for treating breast cancer patients. It is indicated for locally advanced breast cancer after mastectomy and compulsory for those after breast-conserving surgery so as to reduce cancer recurrence and related death [1]. Volumetric modulated arc therapy (VMAT) has emerged as a promising treatment option for this cancer [1,2,3,4,5,6,7]. Compared with conventional conformal techniques, VMAT is capable of achieving superior conformity and homogeneity of target dose and reducing treatment time [4,5,6,8]. However, concerns have been raised regarding dosimetric uncertainty associated with the VMAT mainly because multi-leaf collimator (MLC) patterns used in the VMAT are determined by an inverse planning algorithm, which can only provide optimization for stationary nominal scenarios. As a result, the field aperture can become tightly shaped to the skin surface, which may lead to significant dose loss when chest wall motion even of several millimeters occurs during treatment [9,10,11].

Deep-inspiration breath-hold (DIBH) is a commonly used motion mitigation technique in breast VMAT for reducing radiation dose to the heart. Studies have reported that the DIBH could significantly reduce mean heart dose from 38% to 67% in contrast to the free breathing approach [12,13,14]. Also, the DIBH could enhance target position reproducibility, although residual motion and occasional large chest wall motion may still happen. Hence, the DIBH is not an ideal approach for addressing the dosimetric accuracy issue caused by the chest wall motion. In addition, interfractional breast edema and positioning error can further increase the magnitude of chest wall displacement [14,15,16].

Recently, the notion of plan robustness, which refers to the extent of a radiotherapy plan remaining stable for maintaining the desirable radiation dose even with uncertainties, has emerged in radiotherapy [17,18,19]. Usually, a plan is considered robust when its dosimetric changes are minimal in error scenarios. Currently, manual flash (MF) is the standard robust planning technique in breast VMAT despite its complexity in planning. This technique involves intentionally expanding the planning target volume (PTV) contour beyond the patient’s body for creating a pseudo-structure which is treated as part of the body’s external contour for optimization. Subsequently, the MLCs are expanded further away from skin flash to achieve robust planning. It is reported that the MF plan could improve the target coverage without any significant increase in dose to organs at risk (OARs) [2,20]. 

Nonetheless, robust optimization (RO) originated from proton therapy for range uncertainty compensation that may simplify the planning workflow has been suggested as a better approach to achieve robust planning in VMAT [21,22,23]. This approach requires defining a magnitude of uncertainties and discretizing them into several error scenarios for generating a plan that can minimize objective functions based on the worst case scenarios. In this way, a skin flashing effect could be achieved [24,25]. 

Recent studies have explored the use of RO planning for breast cancer. For example, Jensen et al. [26] compared the free-breathing RO VMAT with DIBH conformal radiotherapy. Dunlop et al. [27] investigated the application of RO planning for addressing the organ deformation issue. Liang et al. [25] evaluated the robustness of the MF and RO approaches and reported that the RO planning was more robust but highlighted the necessity of further studies on this due to small sample size of five patients in their feasibility study. The purpose of this study was to compare the plan robustness of RO and MF approaches in terms of dose variation to clinical target volume (CTV) and OARs. We hypothesized that the RO approach was more robust than the MF technique.

## 2. Materials and Methods

### 2.1. Patient Selection and Simulation

This was a retrospective study involving 34 female patients of Pamela Youde Nethersole Eastern Hospital in Hong Kong Special Administrative Region treated with VMAT for breast cancer between January 2020 and December 2021. Patient inclusion criteria were (1) breast-conserving surgery/total mastectomy for unilateral breast cancer performed, (2) whole breast/chest wall and regional lymph node irradiation prescribed, and (3) DIBH VMAT received [22,25]. There was no restriction on the involved side, staging, and histology. However, certain patient exclusion criteria were applied to ensure the dataset integrity, which were (1) incomplete medical records and (2) inadequate imaging data for VMAT planning. Table 1 shows the patient characteristics. CT Big Bore (Koninklijke Philips N.V., Amsterdam, The Netherlands) was used for simulation computed tomography (CT) scans with the patients positioned in a supine position on a vacuum bag, a slice thickness of 3 mm, and breath hold as per the routine protocol of Pamela Youde Nethersole Eastern Hospital. This study was conducted in accordance with the Declaration of Helsinki and approved by the Institutional Review Board of The Hong Kong Polytechnic University (approval number: HSEARS20230727003 and date of approval: 3 August 2023) and Research Ethics Committee of Hong Kong East Cluster of Hospital Authority of Government of Hong Kong Special Administrative Region (approval number: HKECREC-2022-055 and date of approval: 29 September 2022). Patient consent was waived due to the retrospective nature [7].

### 2.2. Targets and OARs Segmentation

The 34 simulation CT datasets of the selected patients in Digital Imaging and Communications in Medicine (DICOM) format were imported from the CT Big Bore simulator to Eclipse treatment planning system (Varian Medical Systems, Palo Alto, CA, USA) for CTV, PTV, and OARs segmentation in breast VMAT. CTV and PTV were contoured by a radiation therapist with more than 10 years of experience as per the international guidelines. Primary CTV (CTV_p_) was defined as all apparent glandular breast tissue of the involved breast or chest wall. The nodal CTV (CTV_n_) included internal mammary nodes (IMNs), supra-clavicular fossa (SCF), and selected axillary nodes. CTV_p_ and CTV_n_ were combined into a single CTV structure for treatment planning. PTV was derived through expanding the CTV by 8 mm isotropically, with anterior expansion outside the body contour being excluded from the expanded structure. OARs included heart, ipsilateral lung, contralateral lung, and spinal cord [28,29].

### 2.3. Treatment Planning

One MF and one RO plan was generated for each planning CT dataset (imported from Eclipse treatment planning system) on RayStation 12A treatment planning system (RaySearch Laboratories AB, Stockholm, Sweden) with TrueBeam linear accelerator (Varian Medical Systems, Inc., Palo Alto, CA, USA) and a minimum MLC width of 2.5 mm was selected. A 6 MV quarter arc was used in both plans with collimator angles specific to individual patients’ conditions for achieving optimal cardiac sparing [30]. However, the collimator angle of each arc in every MF plan was consistent with the respective RO plan. The prescription was standardized to deliver 50 Gy in 25 fractions without simultaneous integrated boost. The plans were optimized using a maximum of 40 iterations per optimization cycle and the final dose was calculated using a collapsed cone [31].

The planning goal of both MF and RO plans was based on the RTOG 1304/NSABP B51 protocol [32]. However, for maximum dose to PTV, Pamela Youde Nethersole Eastern Hospital protocol, which required that the maximum dose not exceeding 113% of the prescribed dose, was followed. The MF and RO plans were devised based on the PTV and CTV, respectively. Nonetheless, both approaches had the same OAR planning goals [32]. Table 2 shows the detailed dose criteria.

The MF approach used in this study required creation of a virtual bolus and a pseudo-PTV. The virtual bolus was created by adding a tissue-equivalent bolus structure with a thickness of 2 cm covering the entire affected breast, while the pseudo-PTV was generated through expanding the PTV beyond the skin in the anterior and lateral direction by 2 cm [2]. Soft-tissue equivalent Hounsfield unit was assigned to the pseudo structures, which were included in the optimization. The virtual bolus was linked to each arc, and objectives for pseudo-PTV were set. These structures were removed before the dose calculation. The planning strategy enabled MLCs to open away from the external body, with adequate margins to minimize the dose uncertainty because of the chest wall displacement [33,34].

For the RO planning to address the uncertainties, including the residual chest wall motion during breath-hold radiotherapy, breast swelling, and positioning error, 1 cm isotropic patient position uncertainty was employed [34]. Effects of these uncertainties were investigated through generation of 21 error scenarios. Planning with respect to the CTV focused on two key aspects: the skin flashing capability and the robustness of RO plan to address the dose variations in the error scenarios without any manual creation of the pseudo structures [23,35]. Every MF-RO plan pair had the same isocenter. Serial parameter optimization and dose calculation (40 iterations) were performed until clinical goals were met. Dose distribution examples of the MF and RO plans are illustrated in Figure 1. The planning procedure of the MF and RO approaches is summarized in Figure 2.

### 2.4. Plan Evaluation

For evaluation of the MF and RO plan robustness, the simulated organ motion tool available on the RayStation treatment planning system was utilized. This algorithm deformed non-CTV structures with respect to CTV motion as per the user’s maximum magnitude input [36]. In this study, 1 cm isotropic organ motion uncertainty was used, resulting in 21 CT datasets with different combinations of CTV shifting. The dose calculations were performed for all deformed CT datasets of the plans. The plan robustness was quantified by measuring volume/dose difference between nominal and worst scenarios (ΔV/ΔD) for each concerned parameter. The parameters used in this study were based on similar dosimetric studies. This evaluation method could address dose degradation in the worst possible error scenario in line with the definition of robustness. A smaller difference represented greater robustness [24,25]. 

### 2.5. Statistical Analysis

SPSS Statistics 28 (International Business Machines Corporation, Armonk, NY, USA) was used for statistical analysis. Mean and standard deviation were calculated for nominal dose statistics and dosimetric variation for the CTV and OARs of the MF and RO plans. Paired sample *t*-test was used to compare the aforementioned MF plan robustness values with those of the RO plans. A *p*-value less than 0.05 represented statistical significance [37,38,39,40,41].

## 3. Results

Nominal dose statistics of the CTV and OARs of the MF and RO plans are shown in Table 3. Although both MF and RO plans were optimized, the RO plans had statistically significantly smaller mean values for CTV V_95%_, CTV V_100%_, CTV D_98%_, and contralateral lung V_500cGy_ in the normal scenario (*p* < 0.001–0.002). The skin flashing effects achieved in the MF and RO plans are illustrated in Figure 3.

Table 4 demonstrates the dosimetric variation to the CTV and OARs of the MF and RO plans expressed in ΔV/ΔD. All parameters (except CTV ΔD_98%_) of the RO approach had smaller ΔV/ΔD values than those of the MF. Also, the RO approach had statistically significantly smaller ΔV/ΔD values (*p* < 0.001–0.012) for all CTV parameters, except the CTV ΔV_95%_ and ΔD_98%_ and heart ΔD_mean_. Smaller variations (standard deviations) of ΔV/ΔD of all parameters (excluding CTV ΔD_98%_) of the RO approach are noted as well. Figure 4 and Figure 5 show the box-and-whisker plots of the selected CTV and OARs parameters, respectively. Findings of these figures were in line with those of Table 4, including comparable central tendency (mean/median) values between the MF and RO approaches for the CTV ΔV_95%_, ipsilateral and contralateral lung metrics, ΔV_2000cGy_ and ΔV_500cGy_, and less robust performance of the RO for the CTV ΔD_98%_. The results of Table 4 and Figure 4 and Figure 5 indicate that the RO approach was more robust overall.

Figure 6 illustrates dose–volume histograms (DVHs) for doses to tissues in all simulated scenarios of a case example. Good hotspots control is noted in the error scenario for the RO approach, as evidenced by convergence of green DVH bands in the distal end.

## 4. Discussion

To our best knowledge, this is the first study to compare the plan robustness of RO and MF approaches for breast radiotherapy in terms of dose variation to the CTV and OARs based on the simulated organ motion and a dataset of 34 patients. Hence, it advances the knowledge from Liang et al.’s [25] study on evaluating their robustness based on geometry offset and five patients in 2020. This study’s results confirm Liang et al.’s findings that the RO is more robust than the MF with the use of a larger sample size. Specifically, our findings show that the RO plans outperformed the MF in terms of target coverage, high-dose control, and cardiac sparing. Nevertheless, a weakness of CTV underdose is also noted for the RO approach. Also, both approaches were able to generate clinically acceptable plans (Table 4 and Figure 4 and Figure 5).

Unlike Liang et al.’s [25] study that used the geometry offset approach to evaluate the plan robustness, our simulated organ motion approach appears more comprehensive to reflect complexity of the clinical reality through simulating a range of error scenarios caused by the chest wall motion [27,36,42]. Our study reveals that the CTV ΔV_100%_ of the RO approach was statistically significantly smaller than that of the MF because the difference between the nominal and error cases became more apparent in the MF when the shifting magnitude was greater than 0.5 cm in single or multiple directions (Table 4 and Figure 4). These findings are consistent with those of Liang et al.’s study that unpredictable prescribed dose loss was frequently found when the MLC patterns of MF plans were used for the dose calculation based on the deformed CT datasets. This is because the MF plans were optimized only on a single CT dataset that nonuniform deformation or anatomical movement could not be adequately addressed by the uniform expansion of margin of CTV [25,43].

Another strength of the RO is the better high-dose control for avoiding the common acute side effect of radiotherapy, radiation dermatitis (Table 4). Chen et al. [44] found that the incidence of radiation dermatitis happened more frequently when the volume receiving 107% of the prescribed dose within the PTV increased. In our study, the CTV volume receiving more than 107% of the prescribed dose was minimized during planning. Our results reveal that the RO plans were generally more effective than the MF approach in minimizing the unwanted hotspots in the error scenarios. The robustness setting in the RO plans was linked to the objectives for the high dose control, allowing the optimizer to avoid the appearance of the unwanted high doses within the CTV in all scenarios. In contrast, the MF planning had difficulty in eliminating the high doses, as regions of extremely high dose, sometimes up to 120% of the prescribed dose, are observed in the worst scenario of some MF plans. This indicates that its changes in the dose distribution in the error scenarios were unstable. Our results are consistent with those of Dunlop et al.’s study about the robustness evaluation on robust and non-robust planning techniques for breast and internal mammary chain radiotherapy. They indicated that the unpredictable hotspot appearance in the MF planning was due to the use of sophisticated MLC patterns and fluences generated from the optimization based on the nominal scenario only. This resulted in a statistically significantly greater CTV ΔD_max_ for the MF approach [27].

This study’s results also show that the RO approach had better heart dose sparing performance than the MF, as evidenced by its statistically significantly smaller heart ΔD_mean_ value (Table 4). Similar findings are noted in Mahmoudzadeh et al.’s [45] and Chau et al.’s [46] studies on robust and non-robust planning techniques and these recommended the use of robust planning for any breathing control mode. This is attributed to the fact that the RO approach ensures the radiation dose being more conformed to the target, instead of unintentionally falling on the heart, during chest wall motion due to breathing [45]. Usually, an improved target coverage in breast VMAT planning can result in a larger volume of surrounding tissues being irradiated with a low dose, leading to undesirable side effects of low-dose baths on the lungs and heart [45]. Although the RO was better than the MF for cardiac sparing in the worst scenario, both approaches had similar performances for sparing lung doses (Table 4). Again, our findings match those of other studies by Dunlop et al. [27] and Hongo et al. [47]. 

Nonetheless, the MF approach performed better than the RO in terms of minimizing CTV underdose in the worst scenario, as indicated by the statistically significantly smaller CTV ΔD_98%_ value of the MF (Table 4). In contrast, Byrne et al.’s [18] study on the effect of setup variation in breast radiotherapy planning demonstrated that both MF and RO could achieve comparable robustness. This difference can be attributed to the variation in the evaluation approach. However, our findings also illustrate that both MF and RO plans were clinically acceptable. The MF approach is the standard breast VMAT planning technique. Previous studies reported that the MF plans were more robust than the no-flash VMAT plans for the whole breast irradiation due to the achievement of a skin flashing effect [2,31]. Figure 6 confirms that the RO could achieve a similar skin flashing effect but without any need for manual creation of pseudo skin flash, which could result in potential time saving despite not being the focus of this study. For the MF planning, the anatomy for inverse planning optimization is different from the actual anatomy for the dose calculation. Hennet et al. [30] indicated that this is a convergence error because the optimization result converges to an optimal solution based on inaccurate anatomy. The choice of virtual bolus thickness and magnitude of PTV expansion might also influence the plan robustness [48]. The optimal thickness of the virtual bolus is controversial, as a thicker bolus could deteriorate the plan robustness, while a thinner bolus may not create enough skin flash region to compensate the chest wall motion. In contrast, the RO approach did not require any pseudo structure handling, increasing its robustness overall [49].

This study had two major limitations. The deformed CT datasets used might not fully replicate the actual complex chest wall displacement of patients during the treatment. Chau et al. [46] suggested the use of four-dimensional cone beam CT (4D-CBCT) for better simulating the target motion amplitude and direction. Although the 4D-CBCT was not used in this study, our method was recommended by other studies, as it could provide more accurate simulation than the approach involving rigid isocenter shifts [27,33]. Also, equal probability of occurrence was assigned to each scenario, which might not reflect the clinical reality. However, this was the intrinsic limitation of the robust planning module of the RayStation planning system [36].

## 5. Conclusions

This study’s results show that the RO approach was more robust than the MF technique in general. Specifically, the RO plans outperformed the MF in terms of target coverage, high-dose control, and cardiac sparing but with the CTV underdose weakness. Although both RO and MF plans were clinically acceptable, the RO approach is a simpler process, potentially increasing the workflow efficiency. Hence, it is expected that more clinical centers will adopt the RO approach in the future due to the potential improvements of plan robustness and workflow. Further studies should focus on evaluation of the potential time-saving benefit of the RO technique as well as the use of 4D-CBCT for the simulation of target motion amplitude and direction in the plan robustness evaluation.

## Figures and Tables

**Figure 1 diagnostics-13-03395-f001:**
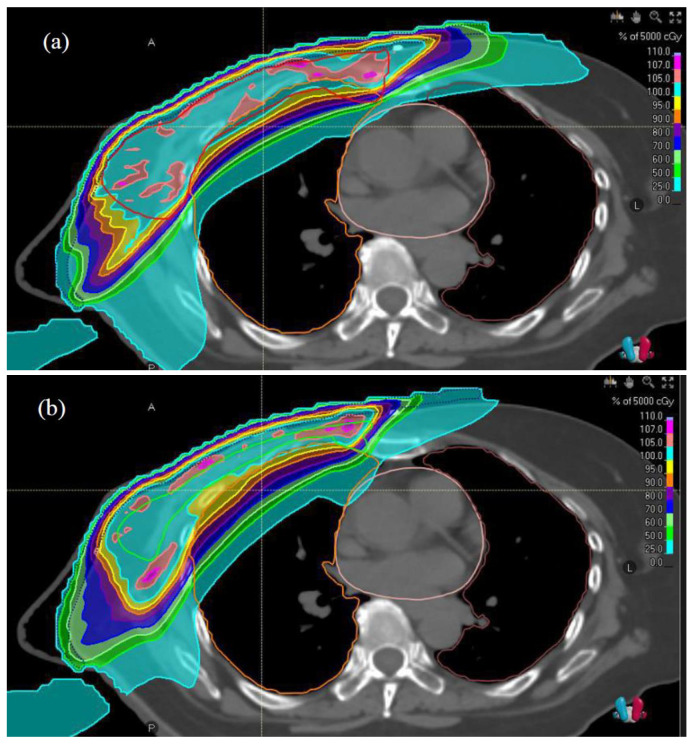
Isodose distributions on principal plane of (**a**) manual flash and (**b**) robust optimized plans for same patient. Red solid line represents contour of planning target volume and green solid line shows contour of clinical target volume. A, anterior; L, left; P, posterior.

**Figure 2 diagnostics-13-03395-f002:**
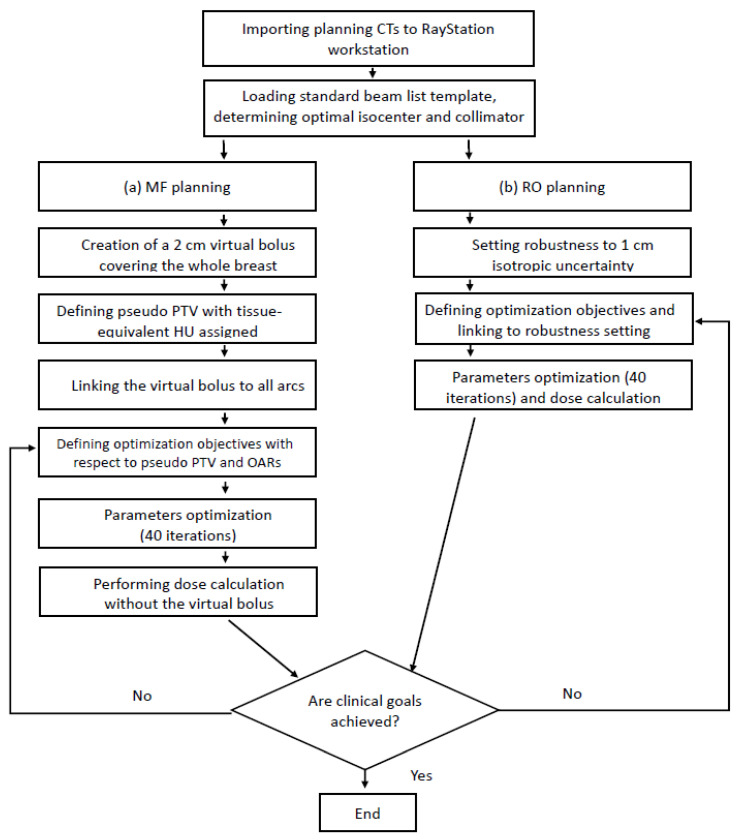
Procedure for (a) manual flash (MF) and (b) robust optimized (RO) plannings. CT, computed tomography; HU, Hounsfield unit; OARs, organs at risk; PTV, planning target volume.

**Figure 3 diagnostics-13-03395-f003:**
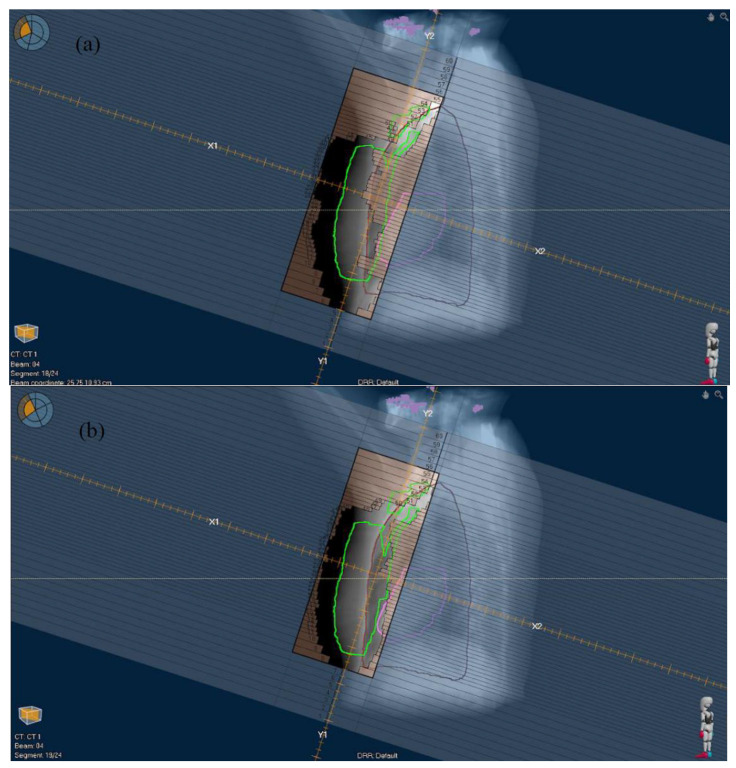
Beam’s eye view shows one beam segment in direction tangent to breast surface for (**a**) manual flash and (**b**) robust optimized plans. Clinical target volume is indicated by green contour.

**Figure 4 diagnostics-13-03395-f004:**
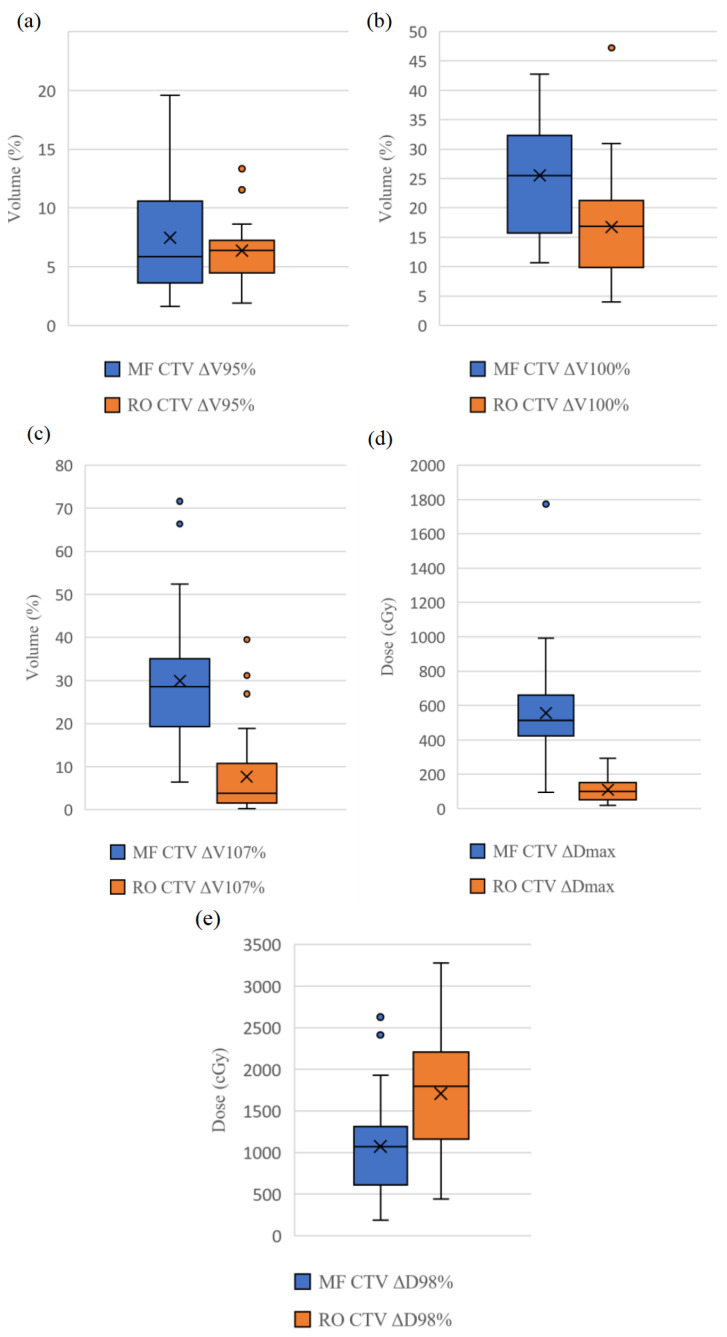
Box-and-whisker plots show dose/volume difference between nominal case and worst case (ΔV/ΔD) for parameters (**a**) CTV V_95%_, (**b**) CTV V_100%_, (**c**) CTV V_107%_, (**d**) CTV D_max_, and (**e**) CTV D_98%_. CTV, clinical target volume; D_98%_, dose received by 98% of structure; D_max_, maximum dose received by structure; MF, manual flash; RO, robust optimized; V_95%/100%/107%_, volume of structure receiving more than 95%/100%/107% of prescribed dose, respectively.

**Figure 5 diagnostics-13-03395-f005:**
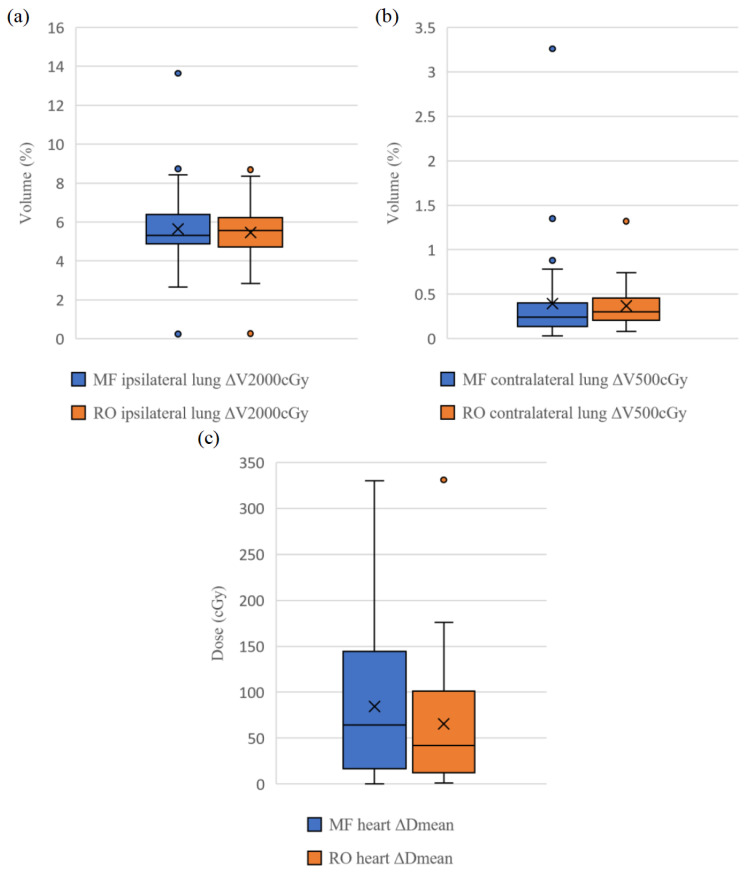
Box-and-whisker plots show dose/volume difference between nominal case and worst case (ΔV/ΔD) for parameters (**a**) ipsilateral lung V_2000cGy_, (**b**) contralateral lung V_500cGy_, and (**c**) heart D_mean_. D_mean_, average dose received by structure; MF, manual flash; RO, robust optimized; V_500/2000cGy_, volume of structure receiving 500/2000 cGy dose, respectively.

**Figure 6 diagnostics-13-03395-f006:**
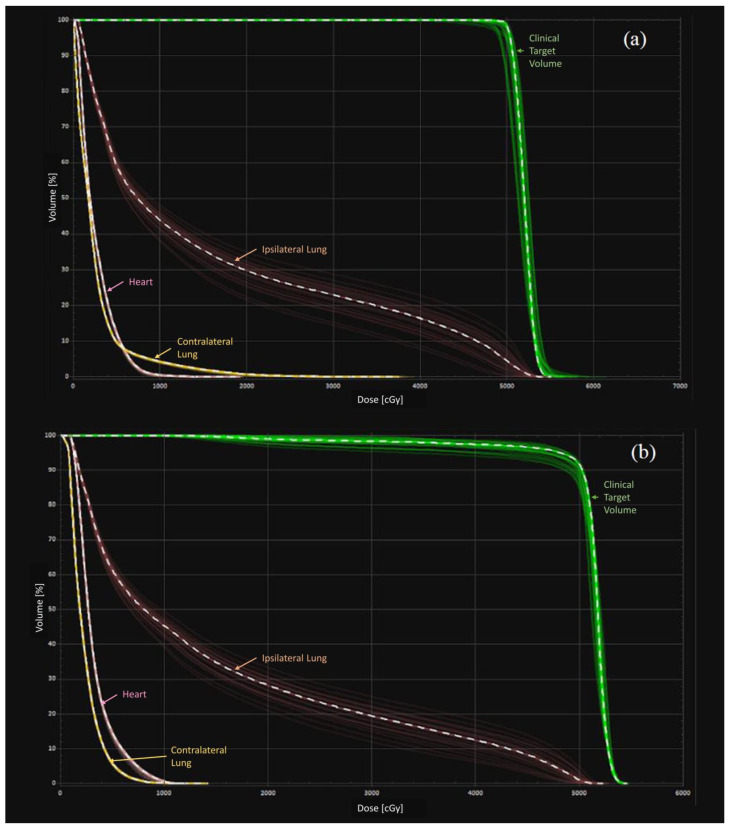
Dose–volume histograms (DVHs) of (**a**) manual flash and (**b**) robust optimized plans.

**Table 1 diagnostics-13-03395-t001:** Patient characteristics (*n* = 34).

Characteristics	Value
*Age*	
<50 years	6 (17.6%)
50–65 years	17 (50.0%)
>65 years	11 (32.4%)
*Involved side*	
Left	22 (64.7%)
Right	12 (35.3%)
*Previous surgical treatment*	
Breast-conserving surgery	6 (17.6%)
Mastectomy	28 (82.4%)
*Overall staging*	
I	3 (8.8%)
II	9 (26.5%)
III	19 (55.9%)
IV	3 (8.8%)
*Histology*	
Invasive ductal carcinoma	21 (61.8%)
Papillary carcinoma	1 (2.9%)
Lobular carcinoma	3 (8.8%)
Other (unknown)	9 (26.5%)

Figures in parentheses are proportions.

**Table 2 diagnostics-13-03395-t002:** Dose constraints for breast volumetric modulated arc therapy planning.

	Parameter	1st Criteria	2nd Criteria
**Target Volume**	PTV (for manual flash plan) V_95%_	≥95%	-
	CTV (for robust optimized plan) V_95%_	≥95%	-
	PTV (for manual flash plan) V_100%_	≥90%	-
	CTV (for robust optimized plan) V_100%_	≥90%	-
**Organs at Risk**	Lung (ipsilateral) V_500cGy_	≤65%	≤70%
	Lung (ipsilateral) V_1000cGy_	≤50%	≤60%
	Lung (ipsilateral) V_2000cGy_	≤30%	≤35%
	Lung (contralateral) V_500cGy_	≤10%	≤15%
	Heart (for left breast cancer) V_1500cGy_	≤30%	≤35%
	Heart (for left breast cancer) V_2500cGy_	≤5%	-
	Heart (for left breast cancer) V_3000cGy_	-	≤5%
	Heart D_mean_	≤400 cGy	≤500 cGy

CTV, clinical target volume; D_mean_, average dose received by structure; PTV, planning target volume; V_95%/100%_, volume of structure receiving more than 95%/100% of prescribed dose, respectively; V_500/1000/1500/2000/2500/3000cGy_, volume of structure receiving 500/1000/1500/2000/2500/3000 cGy dose, respectively.

**Table 3 diagnostics-13-03395-t003:** Nominal dose statistics for manual flash (MF) and robust optimized (RO) plans.

	Parameter	MF Plan	RO Plan	*p*-Value
**Target volume**	CTV V_95%_ (%)	99.61 ± 0.07	97.67 ± 0.21	<0.001 *
	CTV V_100%_ (%)	97.45 ± 0.25	92.51 ± 0.43	<0.001 *
	CTV V_107%_ (%)	5.35 ± 0.70	4.08 ± 0.33	0.105
	CTV D_98%_ (cGy)	4995.00 ± 6.41	4466.62 ± 101.17	<0.001 *
	CTV D_2%_ (cGy)	5382.38 ± 5.83	5377.06 ± 4.39	0.484
	CTV D_max_ (cGy)	5509.03 ± 10.64	5523.94 ± 10.79	0.326
**Organs at risk**	Ipsilateral lung V_500cGy_ (%)	60.64 ± 0.56	60.97 ± 1.01	0.809
	Ipsilateral lung V_1000cGy_ (%)	44.44 ± 0.44	44.50 ± 0.44	0.921
	Ipsilateral lung V_2000cGy_ (%)	30.02 ± 0.42	29.78 ± 0.71	0.768
	Contralateral lung V_500cGy_ (%)	9.58 ± 0.33	8.08 ± 0.42	0.002 *
	Heart D_mean_ (cGy)	370.00 ± 12.58	345.76 ± 10.85	0.114

Figures are expressed in mean ± standard deviation. CTV, clinical target volume; D_2%/98%_, dose received by 2%/98% of structure, respectively; D_max_, maximum dose received by structure; D_mean_, average dose received by structure; V_95%/100%/107%_, volume of structure receiving more than 95%/100%/107% of prescribed dose, respectively; V_500/1000/2000cGy_, volume of structure receiving 500/1000/2000 cGy dose, respectively. * denotes statistically significant difference.

**Table 4 diagnostics-13-03395-t004:** Dosimetric variation to clinical target volume (CTV) and organs at risk of manual flash (MF) and robust optimized (RO) plans.

	Parameter	MF Plan	RO Plan	*p*-Value
**Target Volume**	CTV ΔV_95%_ (%)	7.46 ± 0.88	6.37 ± 0.47	0.197
	CTV ΔV_100%_ (%)	25.56 ± 1.61	16.71 ± 1.50	<0.001 *
	CTV ΔV_107%_ (%)	29.88 ± 2.47	7.63 ± 1.60	<0.001 *
	CTV ΔD_98%_ (cGy)	1071.18 ± 103.09	1707.79 ± 121.26	<0.001 *
	CTV ΔD_2%_ (cGy)	249.44 ± 18.1	58.26 ± 7.27	<0.001 *
	CTV ΔD_max_ (cGy)	556.50 ± 49.69	109.85 ± 10.99	<0.001 *
**Organs at Risk**	Ipsilateral lung ΔV_500cGy_ (%)	3.30 ± 0.27	3.16 ± 0.19	0.570
	Ipsilateral lung ΔV_1000cGy_ (%)	4.29 ± 0.24	4.22 ± 0.23	0.497
	Ipsilateral lung ΔV_2000cGy_ (%)	5.65 ± 0.37	5.47 ± 0.28	0.472
	Contralateral lung ΔV_500cGy_ (%)	0.39 ± 0.98	0.37 ± 0.42	0.804
	Heart ΔD_mean_ (cGy)	84.23 ± 14.1	65.2 ± 12.1	0.012 *

Figures are absolute volume/dose difference between nominal scenario and worst scenario (ΔV/ΔD), expressed in mean ± standard deviation. D_2%/98%_, dose received by 2%/98% of structure, respectively; D_max_, maximum dose received by structure; D_mean_, average dose received by structure; V_95%/100%/107%_, volume of structure receiving more than 95%/100%/107% of prescribed dose, respectively; V_500/1000/2000cGy_, volume of structure receiving 500/1000/2000 cGy dose, respectively. * denotes statistically significant difference.

## Data Availability

The datasets used in this study are not publicly available due to strict requirements set out by the Research Ethics Committee of Hong Kong East Cluster of Hospital Authority of Government of Hong Kong Special Administrative Region.

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
