# Peer review of "Comparative Study of Plan Robustness for Breast Radiotherapy: Volumetric Modulated Arc Therapy Plans with Robust Optimization versus Manual Flash Approach"

_diagnostics, 2023, doi:10.3390/diagnostics13223395_

Round 1

Reviewer 1 Report

Comments and Suggestions for Authors

I read the manuscript very carefully.

I haven't found any errors.

I find it interesting and suitable for this Journal.

I have only 2 comments.

-In my opinion this manuscript is a "technical note". Evaluate the correct placement.

-Are the results obtained comparable with other literature?

English is good, all paragraphs are well connected.

Comments on the Quality of English Language

Good

Author Response

Reviewer 1

I read the manuscript very carefully.

I haven't found any errors.

I find it interesting and suitable for this Journal.

Response: Thank you for your comments.

I have only 2 comments.

-In my opinion this manuscript is a "technical note". Evaluate the correct placement.

Response: Thank you for your comment. According to the Article Types information available on the journal website (https://www.mdpi.com/about/article_types), "Technical notes are brief articles focused on a new technique, method, or procedure. These should describe important modifications or unique applications for the described method. Technical notes can also be used for describing a new software tool or computational method. The structure should include an Abstract, Keywords, Introduction, Materials and Methods, Results, Discussion, and Conclusions, with a suggested minimum word count of 3000 words." and "[Articles] are original research manuscripts. The work should report scientifically sound experiments and provide a substantial amount of new information. The article should include the most recent and relevant references in the field. The structure should include an Abstract, Keywords, Introduction, Materials and Methods, Results, Discussion, and Conclusions (optional) sections, with a suggested minimum word count of 4000 words. Please refer to the journal webpages for specific instructions and templates." As per the manuscript submission system, our original manuscript word count was 4298, indicating that it is not a brief article (3000 words) and able to meet the Article word count requirement (4000 words). The focus of our manuscript was to compare the plan robustness of robust optimization (RO) and manual flash (MF) approaches. As indicated in the Introduction and Discussion sections of the manuscript, RO and MF are not new techniques, methods, or procedures and Liang et al. [25] already compared these two approaches. However, our study innovations were: 1. using a different evaluation approach, and 2. with a greater sample size than Liang et al.’s study which provided substantial amount of new information. Hence, we believe that our manuscript belongs to the Article type. We hope you will find our response satisfactory.

-Are the results obtained comparable with other literature?

Response: Thank you for your comment. Our obtained results were comparable with other literature including Liang et al.’s [25] study and the details have been given in the Discussion section of the manuscript. We hope you will find our response satisfactory.

English is good, all paragraphs are well connected.

Response: Thank you for your comment.

Reviewer 2 Report

Comments and Suggestions for Authors

Authors have presented an interesting manuscript entitled “Comparative Study of Plan Robustness for Breast Radiotherapy: Volumetric Modulated Arc Therapy Plans with Robust Optimization versus Manual Flash Approach”. The article is original, well structured; easy to read with main emphasis on the detailed analysis of variability in robustness of robust optimized (RO) and manual flash (MF) therapy for breast radiotherapy. Authors have described the concept to a greater extent but the manuscript still need some minor corrections before publishing.

·       Authors are suggested to incorporate some content in introduction section related to significance of breast radiotherapy.

·       Authors are requested to explain the reason for selection of only 5 samples for this study in 2020.

·       Authors are requested to improve the quality of flow chart and explain the same in the MS.

·       Authors are suggested to include a separate section of statistical analysis and explain in more elaborated manner.

·       Authors are suggested to improve the quality and presentation of figures.

·       Conclusion must be improved with inclusion of future prospects.

Comments on the Quality of English Language

Minor English corrections are required.

Author Response

Reviewer 2

Authors have presented an interesting manuscript entitled “Comparative Study of Plan Robustness for Breast Radiotherapy: Volumetric Modulated Arc Therapy Plans with Robust Optimization versus Manual Flash Approach”. The article is original, well structured; easy to read with main emphasis on the detailed analysis of variability in robustness of robust optimized (RO) and manual flash (MF) therapy for breast radiotherapy. Authors have described the concept to a greater extent but the manuscript still need some minor corrections before publishing.

-Authors are suggested to incorporate some content in introduction section related to significance of breast radiotherapy.

Response: Thank you for your comment. The following sentences, “Breast cancer is the most common cancer globally. Breast radiotherapy plays an important role for treating breast cancer patients. It is indicated for locally advanced breast cancer after mastectomy, and compulsory for those after breast-conserving surgery so as to reduce cancer recurrence and related death [1].” have been added to the beginning of the Introduction section for addressing the comment.

-Authors are requested to explain the reason for selection of only 5 samples for this study in 2020.

Response: Thank you for your comment. Liang et al.’s [25] study was a feasibility study on robust optimization for skin flashing. This is also noted in their article title, “Robust optimization for skin flashing in intensity modulated radiation therapy for breast cancer treatment: A feasibility study”. Therefore, only five patients were included in their study. To address this comment, the third last sentence of the Introduction section has been changed to “Liang et al. [25] evaluated the robustness of the MF and RO approaches and reported that the RO planning was more robust but highlighting necessity of further studies on this due to small sample size of five patients in their feasibility study.”

-Authors are requested to improve the quality of flow chart and explain the same in the MS.

Response: Thank you for your comment. For addressing this comment, font and line sizes of Figure 2 have been increased for better visibility and clarity. Also, the following changes have been made to Section 2.3.

The sentence, “One MF and one RO plans were generated for each patient case on RayStation 12A treatment planning system (RaySearch Laboratories AB, Stockholm, Sweden) with TrueBeam linear accelerator (Varian Medical Systems, Inc., Palo Alto, CA, USA) and minimum MLC width of 2.5 mm selected.” has been changed to “One MF and one RO plans were generated for each planning CT dataset (imported from Eclipse treatment planning system) on RayStation 12A treatment planning system (RaySearch Laboratories AB, Stockholm, Sweden) with TrueBeam linear accelerator (Varian Medical Systems, Inc., Palo Alto, CA, USA) and minimum MLC width of 2.5 mm selected.”

Beside, the sentence, “Serial optimization and dose calculation were performed until clinical goals were met.” has been changed to “Serial parameter optimization and dose calculation (40 iterations) were performed until clinical goals were met.”

-Authors are suggested to include a separate section of statistical analysis and explain in more elaborated manner.

Response: Thank you for your comment. A separate section of statistical analysis (2.5. Statistical Analysis) has been included with the following elaborated content, “SPSS Statistics 28 (International Business Machines Corporation, Armonk, NY, USA) was used for statistical analysis. Mean and standard deviation were calculated for nominal dose statistics and dosimetric variation for the CTV and OARs of the MF and RO plans. Paired sample t-test was used to compare the aforementioned MF plan robustness values with those of the RO plans. A p-value less than 0.05 represented statistical significance [37-41].” for addressing this comment.

-Authors are suggested to improve the quality and presentation of figures.

Response: Thank you for your comment. For addressing this comment, the following changes have been made to Figures 2 and 6.

Figure 2: Font and line sizes have been increased for better visibility and clarity.

Figure 6: Font size of the text has been increased and annotations for the curves have been added for better clarity and readability.

-Conclusion must be improved with inclusion of future prospects.

Response: Thank you for your comment. For addressing this comment, the following sentence, “Hence, it is expected that more clinical centers will adopt the RO approach in the future due to the potential improvements of plan robustness and workflow.” has been added to the Conclusions to show the future prospects.

Reviewer 3 Report

Comments and Suggestions for Authors

Dear Authors,

This paper proposes a comparative study between omanual flash and robust optimized breast radiotherapy. I recommend this paper for publication if the authors consider the following corrections. 

1- Abstract no need to show the results.

2- The caption of all figures is supposed to be short. The figure description is better to include in the paragraph.

3- Better to add the research objective of this study.

4-  There are  CT datasets, please give the description 

Comments on the Quality of English Language

Minor 

Author Response

Reviewer 3

This paper proposes a comparative study between manual flash and robust optimized breast radiotherapy. I recommend this paper for publication if the authors consider the following corrections.

1- Abstract no need to show the results.

Response: Thank you for your comment. According to the Instructions for Authors of the Diagnostics journal (https://www.mdpi.com/journal/diagnostics/instructions), “The abstract should be a single paragraph and should follow the style of structured abstracts, but without headings: 1) Background: Place the question addressed in a broad context and highlight the purpose of the study; 2) Methods: Describe briefly the main methods or treatments applied. Include any relevant preregistration numbers, and species and strains of any animals used; 3) Results: Summarize the article's main findings; and 4) Conclusion: Indicate the main conclusions or interpretations.” Hence, results need to be shown in the Abstract for meeting the Diagnostics journal requirement. We hope you will find our response satisfactory.

2- The caption of all figures is supposed to be short. The figure description is better to include in the paragraph.

Response: Thank you for your comment. According to the journal’s Style Guide (https://www.mdpi.com/authors/layout), “captions are obligatory and must be placed above or below objects. They should provide a description of the object such that the reader does not need to refer to the main text to fully understand it. For example

“The four methods used.”

is not helpful to readers, whereas

“The four minimization methods used to find the optimum parameters of the Navier–Stokes equation for three microfluidic devices.”

is better. Recall that figures and captions sometimes appear online separately from the rest of the article and so must make sense when not accompanied by the main text.”

Also, as per the Figure caption requirement of the journal’s Layout Style Guide (https://www.mdpi.com/journal/nanomaterials/awards/1342/download), "Only a one-paragraph caption is allowed."

Hence, figure description needs to be provided in each figure caption for meeting the Diagnostics journal requirement. Also, we have reviewed all figure captions. Each caption is only one paragraph which is in line with the aforementioned journal requirement. We hope you will find our response satisfactory.

3- Better to add the research objective of this study.

Response: Thank you for your comment. The followings are the extracts about the Abstract and Introduction requirements stated in the Instructions for Authors of the Diagnostics journal (https://www.mdpi.com/journal/diagnostics/instructions).

Abstract: “Place the question addressed in a broad context and highlight the purpose of the study.”

Introduction: “It should define the purpose of the work and its significance, including specific hypotheses being tested.”

However, the requirement of stating research objective is not mentioned in the Instructions for Authors of the Diagnostics journal.

We have reviewed our manuscript and the study purpose has been given in the Abstract and Introduction sections for meeting the journal requirements. We hope you will find our response satisfactory.

4-  There are CT datasets, please give the description

Response: Thank you for your comment. For addressing this comment, the following changes have been made.

Section 2.1. Patient Selection and Simulation

The sentence, “CT Big Bore (Koninklijke Philips N.V., Amsterdam, The Netherlands) was used for simulation with the patients positioned in supine position on a vacuum bag, a slice thickness of 3 mm, and breath-hold as per the routine protocol of Pamela Youde Nethersole Eastern Hospital.” has been changed to “CT Big Bore (Koninklijke Philips N.V., Amsterdam, The Netherlands) was used for simulation computed tomography (CT) scans with the patients positioned in supine position on a vacuum bag, a slice thickness of 3 mm, and breath-hold as per the routine protocol of Pamela Youde Nethersole Eastern Hospital.”

Section 2.2. Targets and OARs Segmentation

The sentence, “CTV, PTV and OARs segmentation in breast VMAT was performed on Eclipse treatment planning system (Varian Medical Systems, Palo Alto, CA, USA).” has been changed to “The 34 simulation CT datasets of the selected patients in Digital Imaging and Communications in Medicine (DICOM) format were imported from the CT Big Bore simulator to Eclipse treatment planning system (Varian Medical Systems, Palo Alto, CA, USA) for CTV, PTV and OARs segmentation in breast VMAT.”

2.3. Treatment Planning

The sentence, “One MF and one RO plans were generated for each patient case on RayStation 12A treatment planning system (RaySearch Laboratories AB, Stockholm, Sweden) with TrueBeam linear accelerator (Varian Medical Systems, Inc., Palo Alto, CA, USA) and minimum MLC width of 2.5 mm selected.” has been changed to “One MF and one RO plans were generated for each planning CT dataset (imported from Eclipse treatment planning system) on RayStation 12A treatment planning system (RaySearch Laboratories AB, Stockholm, Sweden) with TrueBeam linear accelerator (Varian Medical Systems, Inc., Palo Alto, CA, USA) and minimum MLC width of 2.5 mm selected.”